



# Microbial community changes induced by Managed Aquifer Recharge activities: Linking hydrogeological and biological processes

Carme Barba[1,2], Albert Folch[1,2], Núria Gaju[3], Xavier Sanchez-Vila[1,2], Marc Carrasquilla[3], Alba Grau-Martínez[4], and Maira Martínez-Alonso[3]

[1]Department of Civil and Environmental Engineering, Universitat Politècnica de Catalunya (UPC), C/Jordi Girona 1-3, 08034 Barcelona, Spain
[2]Associated Unit: Hydrogeology Group (UPC-CSIC)
[3]Department of Genetics and Microbiology, Universitat Autònoma de Barcelona (UAB), 08193 Bellaterra, Spain
[4]Grup de Mineralogia Aplicada i Geoquímica de Fluids, Departament de Mineralogia, Petrologia i Geologia Aplicada, Facultat de Ciències de la Terra, Universitat de Barcelona (UB), C/Martí i Franquès s/n, 08028 Barcelona, Spain
**Correspondence:** Carme Barba (carme.barba@upc.edu)

**Abstract.** Managed Aquifer Recharge (MAR) is a worldwide used technique to increase the availability of water resources. We study how MAR modifies microbial ecosystems, and its implications for enhancing biodegradation processes to eventually improve groundwater quality. We compare soil and groundwater samples taken from a MAR facility located in NE Spain during recharge (with the facility operating continuously for several months) and after four months of no recharge. The study

demonstrates a strong correlation between soil and water microbial prints with respect to sampling location along the mapped infiltration path. In particular, managed recharge practices disrupt groundwater ecosystems by modifying diversity indices and the composition of microbial communities, indicating that infiltration favors the growth of certain populations. Analysis of the genetic profiles showed the presence of nine different bacterial phyla in the facility, revealing high biological diversity at the highest taxonomic range. In fact, the microbial population patterns under recharge conditions agree with the Intermediate

Disturbance Hypothesis. Moreover, DNA sequence analysis of excised DGGE band patterns revealed the existence of indicator species linked to MAR, most notably *Dehalogenimonas sp*, *Nitrospira sp* and *Vogesella sp*. Our real facility multidisciplinary study (hydrological, geochemical and microbial), involving soil and groundwater samples, support that MAR is a naturally-based, passive, and efficient technique with broad implications for the biodegradation of pollutants dissolved in water.

*Copyright statement.* TEXT

# 1   Introduction

As the Intergovernmental Panel on Climate Change has stated for years, climate change is affecting and will continue to affect the availability and quality of freshwater resources, with severe consequences to humans and ecosystems. In particular, the



Mediterranean Basin is expected to become warmer and drier (Bates et al., 2008). Therefore, among other actions, claiming a secure water supply should increase groundwater storage of quality water as a strategic management tool in times of scarcity. Managed Aquifer Recharge (MAR) is a globally used, worldwide extended technology based on refilling aquifers with water from different sources (e.g., river, reclaimed, or opportunity water). MAR facilities are usually intended to recover ground-

water levels or to become water reservoirs, but other objectives can be targeted. It is quite common to take advantage of the potential of soil as a biogeochemical reactor to enhance the quality of water infiltrating the vadose zone, especially in surface replenishment systems (Drewes et al., 2003; Nadav et al., 2012).

The Llobregat River (Catalonia, NE Spain) is fed by about a hundred Waste Water Treatment Plants. While nitrogen, phosphorous and organic matter (COD) are eliminated below the legal limits before treated wastewater is discharged to the river,

emerging organic contaminants (EOCs) are not fully removed (Loos et al., 2013). Consequently, significant concentrations of many EOCs have been detected in the Llobregat River (López-Serna et al., 2012) and its associated groundwater bodies (Jurado et al., 2012).

Biodegradation of EOCs strongly depends on redox conditions (Barbieri et al., 2011; Maeng et al., 2010). In this regard, it has been shown that MAR is a feasible technique capable of partially degrading some of these contaminants (Hellauer et al.,

2017; Massmann et al., 2008), particularly when bioprocesses are enhanced (Grau-Martínez et al., 2018; Schaffer et al., 2015). Infiltration through the soil intrinsically leads to two main consequences in groundwater recharge:

1. Development of different vertical and temporal redox zonations responding to organic matter availability as electron acceptors are consumed (Greskowiak et al., 2006).

2. Development of microbial communities according to the flow paths. Fingering below the surface of the recharge systems

and preferential flow paths in the saturated zone can create anaerobic microsites (e.g. Bridgham et al., 2013) in which oxygen is consumed faster than it can be diffused from oxic zones.

Indeed, MAR implies groundwater quality modifications when compared to natural flow conditions. This includes several parameters such as organic matter, dissolved oxygen content, temperature, pH, electrical conductivity, and nutrients (Rivett et al., 2008; Zhang et al., 2016). Such disturbances have ecological implications, as all these parameters affect the growth and activity

of microorganisms and the corresponding degradation of emerging contaminants (Barbieri et al., 2012; Regnery et al., 2017; Valhondo et al., 2018).

Microbial studies linked to MAR practices involve mostly laboratory experiments (Alidina et al., 2014b; Freixa et al., 2015; Li et al., 2013; Rubol et al., 2014). As for microbial MAR field studies, most relevant research is limited to well injection systems (Ginige et al., 2013; Reed et al., 2008; Zhang et al., 2016) or riverbank filtration conditions (Huang et al., 2015). Onesios-Barry

et al. (2014) compared results from a column experiment and soil samples in a MAR site in the US, focusing on the microbial populations linked to pharmaceutical and personal care products removal, and concluded that microbial composition and structure of both systems were comparable. Regnery et al. (2016) went one step further by relating the relative abundance of functional genes involved in xenobiotic pathways with attenuation of some trace organic chemicals and their byproducts in a combination of laboratory experiments and a full-scale MAR facility. However, in our knowledge, there are no microbial fin-





gerprinting studies of MAR surface infiltration basins, that integrate results from surface water, groundwater and soil samples and comparing them in two different operational periods.

The main goal of this study is to determine how MAR activities induce changes in the microbial communities in a real facility composed of a settling and infiltration pond adjacent to the Llobregat River. We evaluate changes on diversity indices and we incorporate results of the DNA sequence analysis of excised DGGE band patterns for samples taken from different environments and locations within the site and under conditions of recharge and non-recharge. Additionally, we link our results with ecological principles and potential biogeochemical processes (i.e. pollutants degradation) occurring due to MAR activities.

## 2  Material and Methods

### 2.1  The Llobregat MAR site

The Llobregat MAR system is located 15 km inland from the Mediterranean Sea, close to Barcelona city (Figure 1). The aquifer thickness in the vicinity is 10-15 m, with alternating sands and gravels. Non-continuous fine-grained sediments are widely present (Pedretti et al., 2012). The distance from the bottom of the pond to the water table oscillated from 9 m (July 2014) to 7 m (March 2015) in the period under study.

Water enters is diverted from upstream the river to a pre-sedimentation basin. After 2-4 days of residence time, it is diverted to an infiltration basin of 6500 $m^2$. The infiltration capacity has been estimated at 1 $m^3 m^{-2} d^{-1}$ on average, and the local transmissivity of the aquifer is estimated as 14000 $m^2 d^{-1}$ (unpublished).

In 2011, a reactive barrier was installed at the bottom of the infiltration basin to increase the organic load of the infiltration water, and thus promote biological processes through the soil and the vadose zone. The barrier was composed of organic compost (50% in volume) mixed with sand and gravel. Small amounts of clay and iron oxides were added to foster adsorption and ion exchange. Previous studies demonstrated that the reactive barrier enhanced the removal of some emergent contaminants, such as sulfamethoxazole or caffeine (Valhondo et al., 2014). More information about the site and the performance of the reactive barrier can be found in Valhondo et al. (2014).

There are six piezometers distributed in a 500 m transect across the study area (Figure 1). P1, P3, P2, P5 and P10 are fully screened. P8 is a multilevel piezometer drilled at three different depths. Water from piezometer P1 represents background conditions (not affected by recharge).

### 2.2  Hydrochemistry sampling surveys

Two recharge situations were compared to evaluate the effect of MAR on groundwater chemical signature. After six months of continuous recharge operation, a sampling campaign took place in July 2014 (wet campaign). Samples were collected from surface water in both basins and in the existing piezometers at different depths (from -5 to 3 masl, see diamonds in Figure 1).





The second sampling campaign was performed in March 2015 after recharge had been discontinued for four months. In this case, groundwater was also sampled.

Water was analyzed for cations, anions ($Cl^-$, $NO_3^-$, $SO_4^{-2}$, $HCO_3^-$), DOC and TOC. Analytical procedures are widely described in Supplementary Material section.

In both campaigns, temperature and electrical conductivity vertical profiles were mapped along the transect from data obtained at 50-cm intervals (MPS-D8, SEBA Hydrometrie).

### 2.3 Microbial community characterization

Water samples were extracted from the pre-sedimentation and infiltration basins at three locations (entrance, middle, and end)

during recharge conditions (from now on, wet scenario). On the contrary, 3 soil samples were extracted at the same locations in the infiltration basin under non-recharge conditions (termed dry scenario). Soil samples were obtained from around 10 to 50cm in dept. The sampling procedure for soil was done taking into account Lombard et al. (2011) recommendations, especially regarding the variability of microbial communities along a field transect. Soil samples were taken by means of cores, individually disassembled and kept in a sterile bag. Groundwater samples were taken from -5 to 3 masl depending on

the piezometers (10 samples for wet scenario and 7 for dry). All soil and groundwater samples were taken in duplicate, kept in sterile conditions, and preserved in dark at $-4°C$ until being taken to the laboratory for molecular analyses.

Protocols for molecular analyses of liquid and soil samples are thoroughly described in the Supplementary Material.

Once the main microbial communities were characterized, three diversity indices were calculated. The first one is Richness ($S$), defined as the proportional number of microbial species present in a sample, i.e., equal to the total number of bands; the

20 other two, Shannon ($H$), and Evenness ($E$), were calculated for each sample as follows:

$$H = \sum_{i=1}^{S} p_i ln(p_i) \tag{1}$$

$$E = \frac{H}{H_{max}} \qquad ; \text{with } H_{max} = lnS \tag{2}$$

where $p_i$ is the relative intensity of each band of the sample. Values reported correspond to the average of the two replicas.

### 2.4 Soil characterization

To complement the soil microbial community's characterization, particle size measurements of soil samples were taken according to the ASTM guidelines. The soil was sampled in the pre-sedimentation basin and at the entrance, middle, and end of

30 infiltration basin. Soil sampling was performed close to the location where samples were taken for microbial analyses.



## 3 Results

### 3.1 Microbial differences in groundwater linked to recharge conditions

#### 3.1.1 Closing the conceptual flow model

Understanding the flow pattern in MAR basins is essential to explain microbial community dynamics. In this regard, 2D

transects of temperature and conductivity alterations obtained at the time of the sampling campaigns (Figure 2a and Figure 3a) based on vertical profiles indicate that: 1) the vertical flow gradient pushes the existing groundwater downwards and forms a shallow front that travels approximately 120 m downstream, eventually mixing with the background water; 2) the background water is mostly found near the recharge pond and at the deepest sampling points below the pond.

From this conceptual model, four main groups of groundwater can be defined under recharge conditions:

– Type I water represents the background environment of the aquifer, unaffected by MAR activities. Water sampled in P1 is an example of this type.

– Type II water, is the infiltrating one (best observed in P8 at both sampling depths). It flows vertically through the vadose zone to the aquifer, creating a small water mound that pushes down the Type I water.

– Type III water, characteristic of points P2(3), P5(2.3) and P5(-2.2), is a mixture between Types I and II waters, with a

high proportion of the latter.

– Type IV water is again a mixture, but with a lower proportion of Type II water. It is present in piezometers P3(0.8), P3(-4.2), P2(-5) and P10(-1).

Apart from temperature and conductivity, major ions composition does not show any significant trend related with groundwater zonation below the pond (see Table S1 in Supplementary Material). The role of nitrate and DOC in microbial community

patterns is discussed further below.

#### 3.1.2 Clustering groundwater microbial communities according to presence and abundance

To characterize differences in microbial communities due the recharge, groundwater samples were subjected to molecular analysis. Post-processing of DGGE gels allowed for Non-Metric Multidimensional Scaling (NMDS), showing similarities

among band patterns (Figure 4) and strong clustering of microbial communities. Samples from both scenarios were completely separated; blue squares (dry) and triangles (wet) represent groundwater samples, and are clearly clustered in top and bottom halves of the plot, respectively. Moreover, samples from the wet scenario grouped according to water types. Types I and II are displayed on opposite sides; Types III and IV (mixed) display in between. The two green triangles in the center of the plot correspond to groundwater samples from P10.

Discrete bands are also portrayed (circles), allowing linkage of the bands' contribution to sample assemblages. Filled circles



report the class and genus of the sequenced bands, whereas empty circles symbolize non-sequenced bands.

Figure 5 shows DGGE profiles and UPGMA clustering analysis of groundwater samples. The genetic fingerprints revealed high dissimilarities in the bacterial assemblage of about 70% and 80% during the active recharge period and the dry campaign, respectively (Figure 5). Moreover, most replicas grouped together, indicating sampling quality. Under active recharge (wet)

5  conditions (Figure 5a), the dendrogram reproduces quite well the water types postulated by the conceptual flow model: in the first group, we can include four out of the five samples that were strongly influenced by recharge (P5(-2.2), P5(2.3), P8(1) and P8(-3)); while in the second group, P2(3), P3(-4.2) and P3(0.8) clustered together with P1 (non-affected by recharge). In the dry campaign (Figure 5b), although no infiltration occurred, P8 appears separated from the other piezometers, indicating the still marked influence of the water infiltrated during the wet period, which occurred over four months earlier.

### 3.1.3   Variations in microbial diversity indices in groundwater

The structure and processes of ecosystems change when a disturbance occurs (Grimm, 1994). Such changes in microbial communities have been quantified and described by means of diversity indices (Table 1). Such indices, grouped according to water types during wet conditions, were ordered along an imaginary line from low to highly perturbed as a consequence of water

infiltration (Figure 6). The lowest diversity indices were obtained for the recharging water (Type II), indicating low species richness and a highly dissimilar proportion. In contrast, Type IV water, only slightly affected by water infiltration, displayed higher Shannon and Evenness indices, similar to Type I (unaffected by recharge).

### 3.1.4   Role of MAR activities for the microbial community structure

Prominent bands were recovered from the DGGE gels (Figure 5) and sequenced. Table S2 (Supplementary Material) shows the sequenced bands, their similarity values compared to the closest related GenBanK sequences, and their phylogenetic affiliations. Overall, sequences fell into nine different bacterial phyla and eleven classes: Proteobacteria (Alphaproteobacteria, Betaproteobacteria and Gammaproteobacteria), Cyanobacteria, Chloroflexi (Dehalococcoidia), Chlorobi (Chlorobia), Nitrospirae (Nitrospira), Acidobacteria, Actinobacteria, Firmicutes (Bacilli) and Bacteroidetes (Cytophagia) (Figure 7). The group

designated as "Others" includes unclassified and non-sequenced fine bands.

The two main classes displaying the largest differences between the two scenarios are Betaproteobacteria and Dehalococcoidia, which were favored under recharge conditions. In particular, Dehalococcoidia is present in medium- and low-influenced waters, and it is absent in the high recharge-influenced groundwater (P8(1) and P8(-3)). This phylotype was identified at the genus level as *Dehalogenimonas sp* (Table S2). Similar behavior was found in the Nitrospira class, appearing in low-influenced ground-

water in the wet scenario.

Patterns in the structure of microbial populations correlated with water types. For Type I, differences in the bacterial assemblage between both campaigns were attributed to seasonal changes (Table S1). Dehalococcoidia and Chlorobia were only detected in the wet scenario, while Cytophagia and Nitrospira could only be detected under dry conditions.



During the active recharge period and for Type IV water, Dehalococcoidia was found in three out of four sampling points and was the most abundant phylotype. For Type III water, significant differences were observed among in the samples analyzed. Populations with the highest relative abundance in P5 (2.3) were Betaproteobacteria and Bacilli. The former was also prominent in P5(–2.2), together with Cytophagia, while Dehalococcoidia were dominant in P2(3). Finally, in the case of groundwater

5 Type II (recharge water), the bacterial assemblage was dominated by members of the Betaproteobacteria class. During the dry period, no clear distribution patterns in the bacterial relative abundances at the phylum and class level were observed, in part due to the DGGE profiles, mainly composed by fine bands (Figure 5); these were difficult to recover and purify, and thus could not be characterized. However, it should be mentioned that Betaproteobacteria were dominant in both P8 samples, contributing more than 50% to the relative abundance.

## 3.2 Microbial community indicators of MAR in soil and surface water

To study the impact of MAR on microbial community structure, recharge water from pre-sedimentation and infiltration basins, as well as soils, were analyzed. Figures 8 and 9 show the relative abundance of bacterial phylotypes at the taxonomical level of classes for surface water and soil samples. The results are displayed according to the distance to the recharge basin inlet.

Microbial richness in soil samples was controlled by water content. Non-recharge conditions had a primarily negative effect on the populations of Dehalococcoidia, Acidobacteria and Chlorobia, but favored the presence of Nitrospira, Cytophagia and Actinobacteria (Figure 8). Shannon and Evenness indices demonstrated that soils were more diverse under wet conditions than under dry ones (Table 1).

For surface water samples (Figure 9), there was a decreasing gradient in community complexity along the ponds. Acidobacte-
20 ria, Betaproteobacteria and Cyanobacteria were the main phylotypes present.

## 4 Discussion

We contend that interdisciplinary analysis of geochemical characterization, recharge evaluation, and microbial fingerprinting, can provide relevant information about the fate of microbial ecosystems in soil and groundwater.

## 4.1 Matching groundwater model, ecological disturbance principle and microbial communities

Groundwater is a quite stable aquatic environment (Griebler and Lueders, 2009). One could expect that microbial communities in groundwater should also display low variability that could be reflected in the diversity indices. An example of this is P1, which is unaffected by recharge; the diversity indices remain constant regardless of the sampling campaign.





MAR is a passive treatment technique that can provide simultaneously oxic and anoxic conditions (Maeng et al., 2011). This has wide implications for the potential biological removal of selected emerging contaminants, as each micropollutant is most efficiently removed under specific redox conditions (Schmidt et al., 2007). Some can even be degraded by co-metabolism, involving different redox states in the process (Rodriguez-Escales and Sanchez-Vila, 2016). In this sense, MAR is an efficient

remediation system. In addition, many sequenced phylotypes, such as *Nitrospira sp*, *Stenotrophomonas sp* and *Methylobacterium sp* have been associated with degradation capabilities (Cycoń et al., 2017; Daims et al., 2001; Wang et al., 2015). In short, the MAR microbial ecosystem studied in this work presents many more phylotypes than previous studies reported in groundwater systems (Logue et al., 2015), and thus, MAR can be considered an efficient remediation system.

We further tested the Intermediate Disturbance Hypothesis (IDH) for microbial communities in groundwater (Figure 6) related

to MAR activities. IDH was originally proposed for tropical rain forests and coral reefs (Connell, 1978) and supports the idea that small perturbations create new access to resources for species which have overlapping niches, allowing their coexistence. This mechanism, known as a competition-colonization trade-off, can explain IDH in local communities, leading to an increase in diversity. However, when the degree of disturbance rises, only eurytolerant populations can survive and grow. Thus, an inverse correlation between diversity and the degree of disturbance (reflected in the temperature and conductivity profiles) was

expected (see Table 1). Such correlations have also been reported in recharge wells and snowmelt-influenced aquifers (Ginige et al., 2013; Zhou et al., 2012).

In the Llobregat MAR system, the initial diversity in the microbial community increased with perturbation caused by recharge (Figure 6), with maximum diversity associated with Type IV water, and lowest for the most disturbed water (Type II). In ecological terms, Type III and Type IV waters represent different proportions of perturbation.

In the most altered groundwater zone (represented by P8 samples), Betaproteobacteria grew above 50% of the relative abundance (Figure 7). The main phylogenetic affiliation of this phylotype at the genus level is *Vogesella*. Strains of this genus are able to catabolize monosaccharides under aerobic conditions, but not under low-oxygen conditions. Furthermore, all *Vogesella* strains are denitrifiers (Grimes et al., 1997). Indeed, P8, located below the pond, receives oxygen-rich water during the recharge process, driven by fingering in the vadose zone. Although dissolved oxygen was not measured in the present study, data from

other campaigns confirm this behavior for oxygen in P8 samples (data not shown). Moreover, nitrate concentration in the surface water was low (Table S1), and thus most denitrification is expected to occur under the pond. Grau-Martínez et al. (2018) recently confirmed that nitrate was consumed *via* denitrification pathway under the infiltration pond in the Llobregat MAR system, supporting the idea that *Vogesella sp* could be one of the genera involved in nitrate consumption. Likely, depending on the oxygen content, *Vogesella sp* will adapt its metabolic function in favor of aerobic oxidation of organic matter or by means

of denitrification, thus becoming a good indicator of highly-disturbed MAR environments.

For Type III and Type IV waters, *Dehalogenimonas sp*, within the Dehalococcoidia class, is characteristic of medium-disturance groundwater (Figure 4 and Figure 7). *Dehalogenimonas sp* has been studied in recent years because some strains are associated with dechlorination in contaminated sites. This genus is strictly anaerobic and mesophilic, and some species can reductively dehalogenate polychlorinated aliphatic alkanes (Martín-González et al., 2015; Moe et al., 2009). As a result, recharge creates

reducing conditions, likely indicating the existence of microzones or microsites (Bridgham et al., 2013; Hamersley and Howes,





2002), defined as local anoxic areas that coexist with fast-travelling oxygen-rich paths. Thus, microbial analysis can be used to unmask the apparent mishap of water samples that are oxic and display some typical anaerobic species. Moreover, some species of *Dehalogenimonas* can dechlorinate some Trichloroethane isomers (Dillehay et al., 2014), a pollutant reported in the Llobregat Lower Valley at levels as high as 300 $\mu g/L$ (Valhondo et al., 2014), thus opening the door for the development of

enhanced remediation activities.

The Evenness index is an indicator of the equity of a community, and can be quite informative for observing perturbations to microbial communities. In the wet scenario, the lowest values of E were obtained for the samples most affected by recharge (Table 1), indicating that some species developed into predominant members of the microbial assemblage. Groundwater samples displayed the highest evenness values in the area less affected by recharge and in the dry scenario. in the latter case, values

indicate the recovery of microbial communities from the disruption caused by recharge. In fact, P8(1) samples in dry scenario were not fully consistent with this conceptual model, with low evenness index and very low nitrate concentration. Furthermore, the presence of *Methylotenera mobilis* (Betaproteobacteria class) in both P8 sampling points was more than 40%, on average, of the relative abundance. *Methylotenera mobilis* is a methylotroph specie with denitrification abilities (Chistoserdova, 2011). These results suggest that P8 denitrification processes occur below the basin even when it is empty, indicating that four months

is not enough time to revert back to natural conditions at this sampling point. This assumption is consistent with nitrate isotopic data presented in Grau-Martínez et al. (2018), and is also in agreement with the study of Rodriguez-Escales et al. (2016) in which biomass decay acted as an endogenous carbon source for respiration once the input carbon was reduced, maintaining denitrification rates.

**4.2   Microbial community structure in soils and surface waters**

We analyzed the heterogeneity of the microbial community structure in soil and surface water in terms of the distance to the infiltration basin entry point (Figure 9 and Table 1). Patterns in surface water microbial composition are linked to sequential sedimentation processes as revealed by granulometric analyses of soil samples (Table S3). The result was that surface water became poorer in terms of the presence of microbial communities between the pre-sedimentation basin and the end of the

infiltration basin. The main reason could be the decrease in solids suspended throughout the system due to the sequential decantation of particles and their attached biomass. Furthermore, surface water displays relatively higher values of Cyanobacteria and Acidobacteria classes compared to groundwater. Cyanobacteria constitute the largest, most diverse, and most widely distributed group of photosynthetic prokaryotes, which are capable of conducting N fixation (Stanier and Bazine, 1977). However, members of the phylum Acidobacteria are physiologically diverse and ubiquitous in soils, degrade a wide range of carbon

sources (from substances with a wide range of complexity), and are capable of reducing nitrates and nitrites (Kielak et al., 2016). This heterogeneous effect with distance to the entry point of the basin is also observed in the diversity indices, which lose diversity with distance and are inversely correlated to the proportion of fine particles. Similar behavior for richness correlated to soil texture was reported elsewhere (Chau et al., 2011).

Differences in the microbial communities in soils between the two basins were concentrated in the large organic matter content





provided by the reactive barrier present in the latter, being a source for the growth of bacterial communities and enhanced diversity under recharge conditions. The role of the humidity on microbial diversity is also significant, as was previously reported in horizontal subsurface constructed wetlands (Nurk et al., 2005). Furthermore, phylotypes distribution changes among scenarios. Whereas Dehaloccocoidia and Chlorobia classes appear in wet soils, Nitrospira, Cytophagia and Actinobacteria are

favored under dry conditions.

The role of the reactive layer at the infiltration pond could be extrapolated as a system fed with a considerable organic carbon load. Laboratory experiments and constructed wetlands demonstrate that concentration of microbial activity and TOC degradation is concentrated in the first centimeters of the filter material (Ragusa et al., 2004; Sleytr et al., 2007; Tietz et al., 2007) in response to oxygen concentration vertical distribution. Although rapid oxygen depletion and consequent denitrification condi-

tions have been evidenced in lab-scale MAR experiments (Alidina et al., 2014a; Dutta et al., 2015), this effect may not happen rapidly under real infiltration conditions, where entrapped gas (Heilweil et al., 2004) or fingering processes (Kung, 1990) may provide higher oxygen concentrations than in lower dimension systems (e.g., columns). Lab-experiments are doubtlessly useful to elucidate the behavior of microbial communities under controlled conditions. However, sometimes it could exist a difficulty of transferring conclusions obtained from lab samples to real sites.

## 5   Conclusions

This study aims at integrating different fields such as hydrogeology, ecology and microbiology applied to a real MAR facility, relating flow (infiltration) conditions, physicochemical water parameters, and microbial changes induced by managed recharge along vertical transect. We observed that infiltration ponds modify the hydrochemistry and ecology of the groundwater environ-

ment, especially in terms of microbial communities. Comparing recharge and non-recharge scenarios, we found that microbial diversity indices (Shannon) correlate inversely with the degree of perturbation caused by the induced recharge, substantiating an Intermediate Disturbance Hypothesis distribution. In fact, MAR (surface) basins operation can promote different levels of disturbance at the same time, and microbial community structures change accordingly. From microbial fingerprinting analysis, we observed the boosting of Betaproteobacteria and Dehalococcoidia classes correlate to recharge practices. Furthermore,

genera such as *Dehalogenimonas*, *Nitrospira*, *Stenotrophomonas*, *Methylobacterium* were also detected, indicating a wide spectrum of biodegradation capabilities. Likewise, sequencing tasks revealed characteristic phylotypes from each water type, particularly *Vogesella sp* for highly perturbed water or *Dehalogenimonas sp* for medium-perturbed water. Microbial populations in soil are quite diverse when comparing wet with dry scenarios. Soil moisture and sediment grain size appear to be the key factors explaining diversity patterns. Furthermore, variation in recharge conditions do not translate immediately to changes

in communities. All these results combined confirm the difficulty of extending laboratory experiment results to the field scale.





*Competing interests.* The authors declare that they have no conflict of interest.

*Acknowledgements.* This investigation was financially supported by the European Union project MARSOL grant agreement no. 619120, FP7-ENV-2013-WATER-INNO-DEMO, Generalitat de Catalunya via FI scholarship program (FI-DGR 2014) and the Spanish Government and EU (project ACWAPUR PCIN-2015-239). The authors would like to acknowledge Marc Vives for his help and Comunitat d'Usuaris d'Aigües de la Vall Baixa i del Delta del Riu Llobregat (CUADLL) for their cooperation.





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



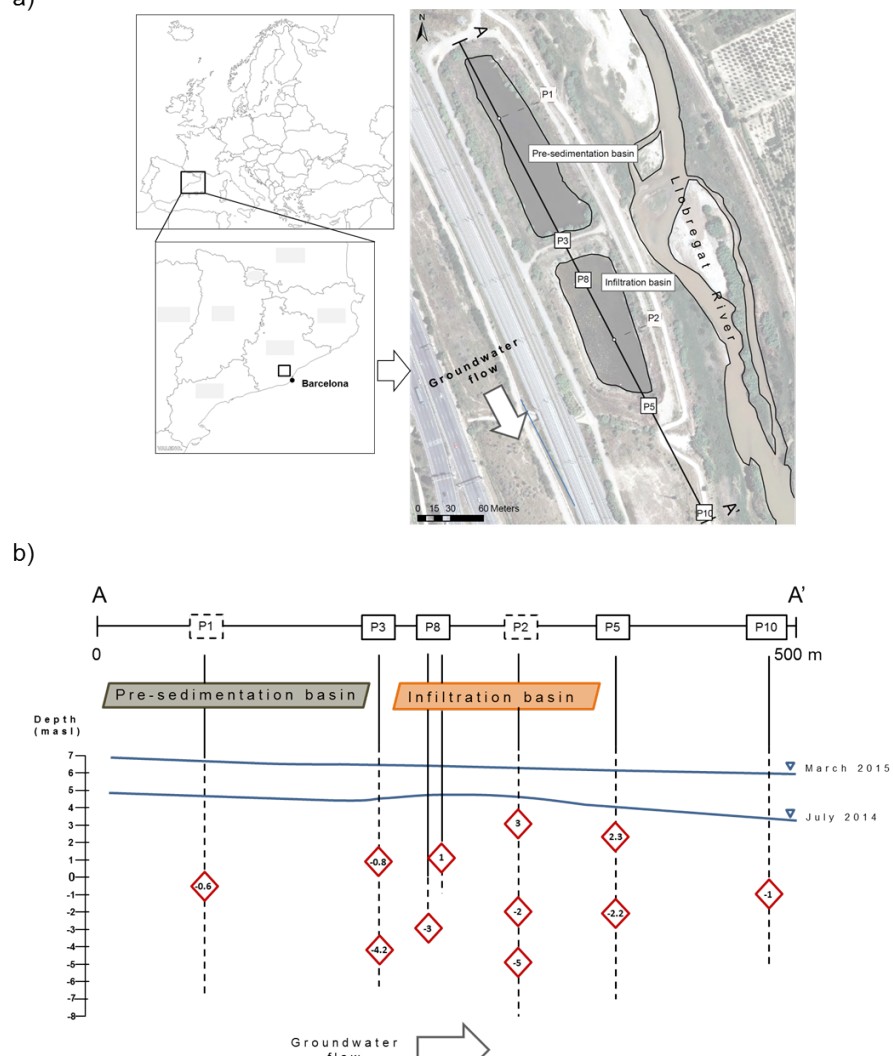

**Figure 1.** Geographical location of the Llobregat MAR system and location of the established transect (a) Transect section with piezometers (P1 and P2 are projected) and displaying sampling depths (red diamonds). (b) Blue line shows groundwater level in July 2014 (wet scenario - recharge) and March 2015 (dry scenario – non-recharge)



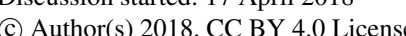


**Figure 2.** Temperature distribution at the local scale in (a) July 2014 (wet), and (b) March 2015 (dry). Red diamonds indicate sampling points for microbial and water analysis in each campaign



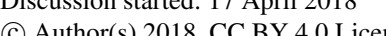

**Figure 3.** Electrical Conductivity distribution at the local scale in (a) July 2014 (wet), and (b) March 2015 (dry)





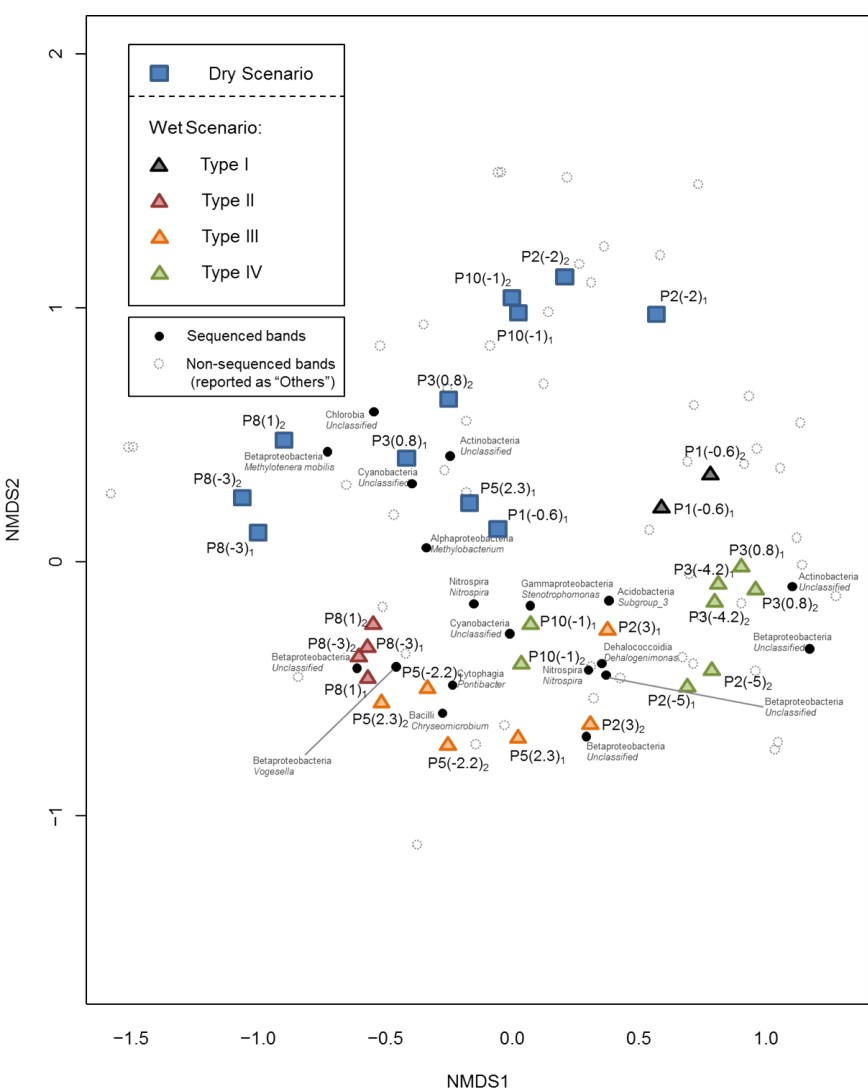

**Figure 4.** Non-metric Multidimensional Scaling clustering for all groundwater samples. Blue squares and triangles represent samples in dry and wet scenarios, respectively. Colors in triangles represent water types. Black circles correspond to band migration numbers in DGGE gels that were sequenced (phylogenetic affiliation corresponding to each black circle). Non-sequenced bands are also portrayed (empty circles)



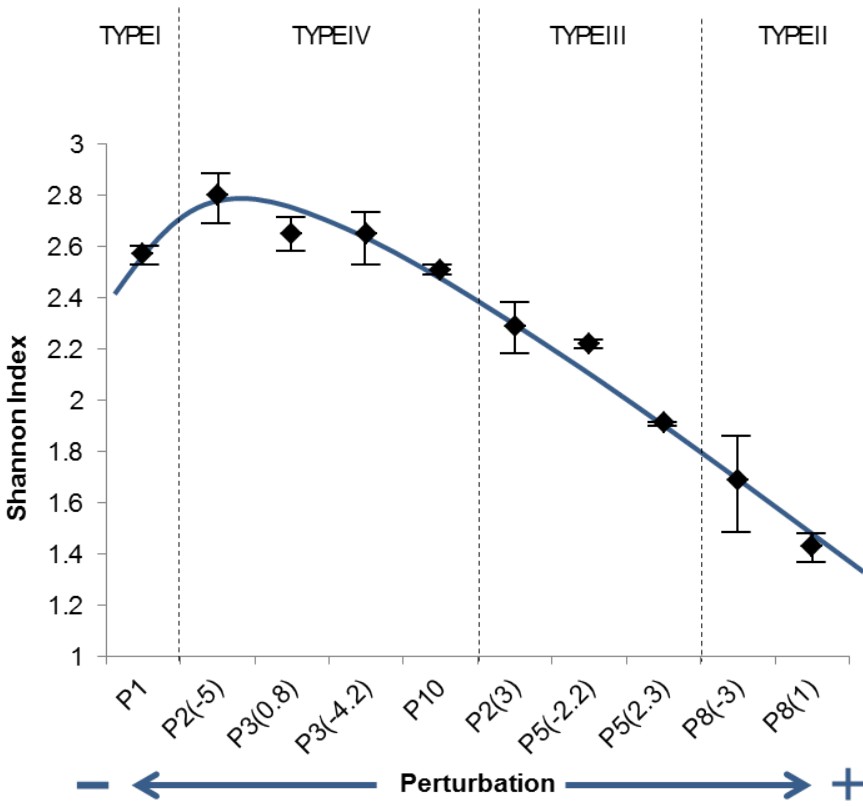

**Figure 5.** Average of Shannon Indices in piezometer samples under wet conditions. Standard deviation is shown in error bars. Horizontal axis reflects the degree of perturbation of original groundwater due to recharge




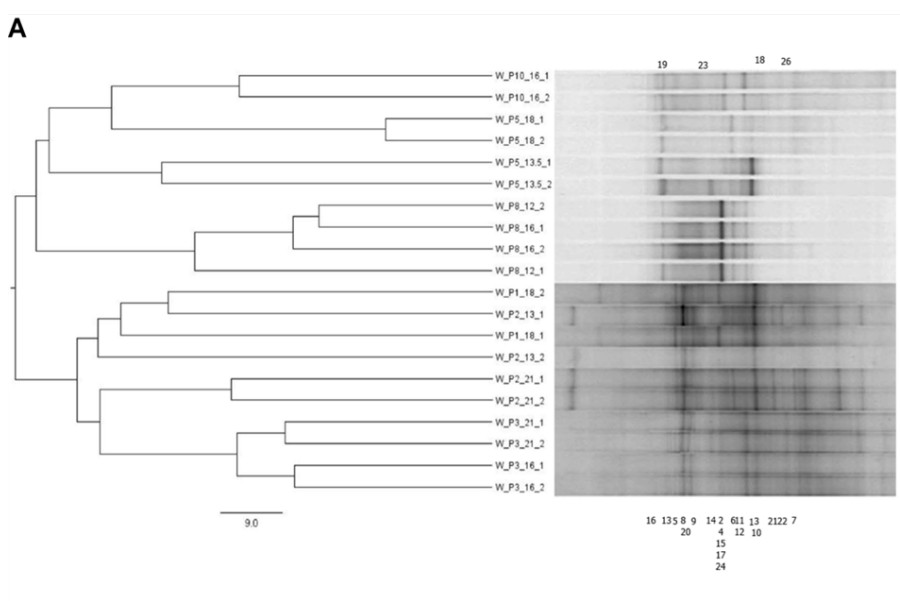

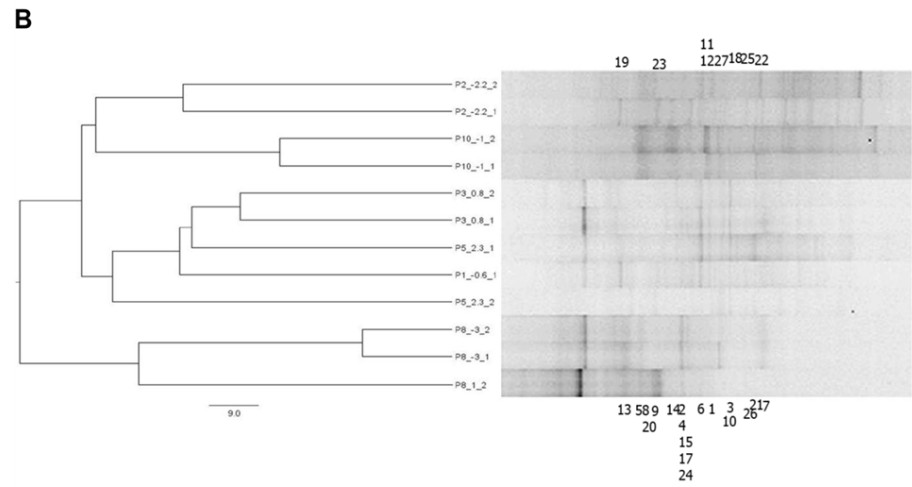

**Figure 6.** DGGE band patterns of bacterial 16S rRNA gene fragments and UPGMA cluster analysis for fingerprints obtained from wet (July 2014) (a) and dry (March 2015) (b) periods. Bar indicates 9% divergence. Each sample is defined by a code indicating piezometer number and sampling depth (see table 1). Black triangles indicate position of bands recovered and sequenced, and numbers correspond to their phylogenetic affiliation (see table S2 supplementary material)



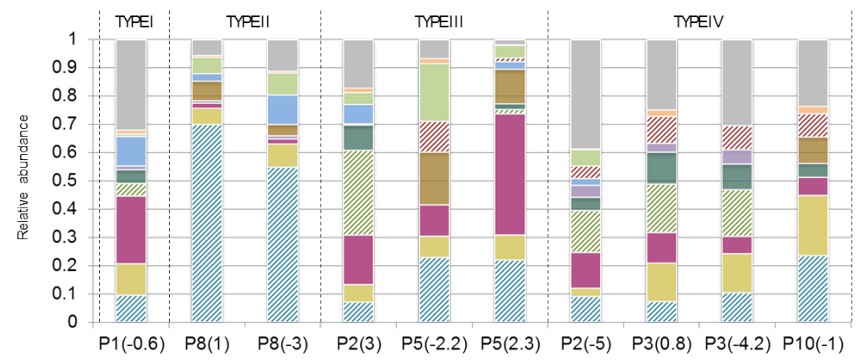

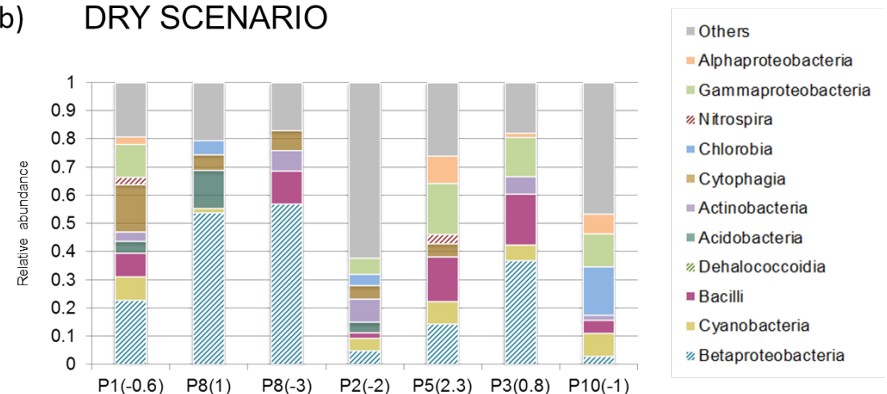

**Figure 7.** Bacterial community structure of groundwater samples. Class relative abundances calculated for wet (a) and dry (b) scenarios





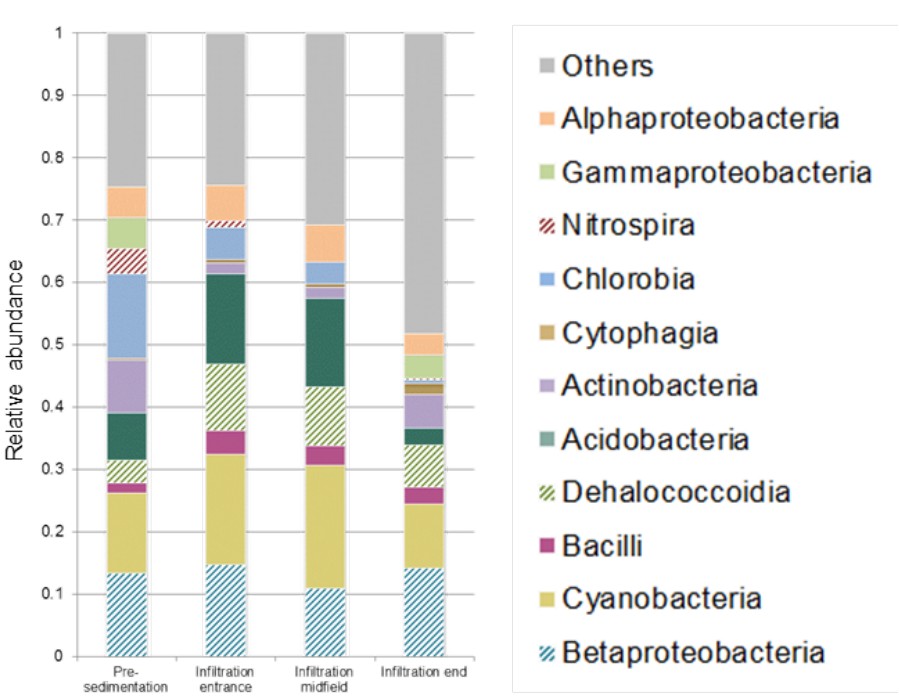

**Figure 8.** Bacterial community structure of soil samples from pre-sedimentation and infiltration basins. Class relative abundances calculated for wet (a) and dry (b) scenarios



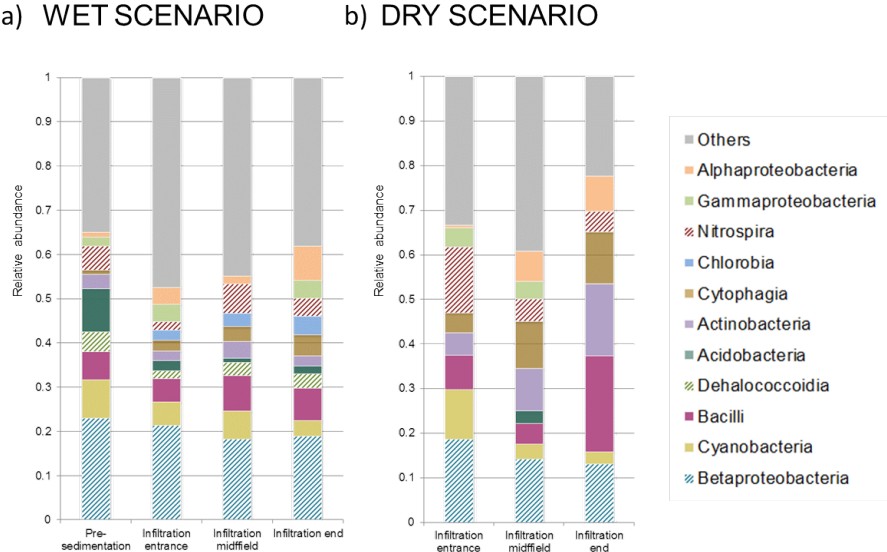

**Figure 9.** Bacterial community structure from water samples of pre-sedimentation and infiltration basins. Class relative abundance calculated for wet scenario





**Table 1.** Summary of values of Shannon, Richness and Evenness indices at the Llobregat MAR site in different scenarios

| Environment | | Sampling location (**depth** -*masl*-) | Shannnon *(SD)*[*] | | Richness *(SD)*[*] | | E *(SD)*[*] | |
|---|---|---|---|---|---|---|---|---|
| | | | WET | DRY | WET | DRY | WET | DRY |
| Groundwater | T.I | P1(**-0.6**) | 2.57 *(0.04)* | 2.73 | 22 *(4.1)* | 19 | 0.61 *(0.01)* | 0.64 |
| | T.II | P8(**1**) | 1.43 *(0.05)* | 1.73 | 10 *(2.0)* | 11 | 0.34 *(0.01)* | 0.41 |
| | | P8(**-3**) | 1.69 *(0.20)* | 2.03 *(0.14)* | 11 | 10 *(2.0)* | 0.40 *(0.05)* | 0.48 *(0.03)* |
| | T.III | P2(**3**) | 2.29 *(0.13)* | | 18.5 *(7.2)* | | 0.54 *(0.03)* | |
| | | P2(**-2**) | | 2.67 *(0.13)* | | 18 *(2.0)* | | 0.63 *(0.03)* |
| | | P5(**2.3**) | 1.91 *(0.01)* | 2.41 | 14.5 *(1.0)* | 14 | 0.45 *($1 \times 10^{-3}$)* | 0.56 |
| | | P5(**-2.2**) | 2.22 *(0.01)* | | 12.5 *(1.0)* | | 0.52 *($2 \times 10^{-3}$)* | |
| | T.IV | P3(**0.8**) | 2.65 *(0.09)* | 2.01 *(0.28)* | 21 | 10.5 *(3.07)* | 0.63 *(0.02)* | 0.48 *(0.07)* |
| | | P3(**-4.2**) | 2.65 *(0.13)* | | 20 | | 0.63 *(0.03)* | |
| | | P2(**-5**) | 2.80 *(0.12)* | | 23 *(4.1)* | | 0.66 *(0.12)* | |
| | | P10(**-1**) | 2.51 *(0.20)* | 2.49 *(0.09)* | 17 | 16 *(4.1)* | 0.59 *(0.01)* | 0.59 *(0.02)* |
| Water of basins | | Pre-sedimentation | 2.93 *(0.28)* | | 29 *(14.3)* | | 0.69 *(0.07)* | |
| | | Infiltration entrance | 2.78 *(0.53)* | | 28.5 *(17.4)* | | 0.66 *(0.12)* | |
| | | Infiltration midfield | 2.66 *(0.14)* | | 25.5 *(3.1)* | | 0.63 *(0.03)* | |
| | | Infiltration end | 2.58 *(0.08)* | | 24 *(2.0)* | | 0.61 *(0.02)* | |
| Soil of basins | | Pre-sedimentation | 2.89 *(0.16)* | | 25.5 *(7.2)* | | 0.68 *(0.04)* | |
| | | Infiltration entrance | 3.22 *(0.09)* | 2.93 *(0.21)* | 35.5 *(1.0)* | 25 *(7.6)* | 0.76 *(0.02)* | 0.69 *(0.08)* |
| | | Infiltration midfield | 3.36 *(0.14)* | 2.90 *(0.05)* | 30.5 *(3.1)* | 21 | 0.79 *(0.03)* | 0.68 *(0.01)* |
| | | Infiltration end | 3.29 *(0.05)* | 2.40 *(0.07)* | 34 *(2.0)* | 16 | 0.78 *(0.01)* | 0.57 *(0.02)* |

[*] Numbers in parenthesis after each index value indicate standard deviation, not reported whenever one replica was damaged.