# Peer review of "Microbial community changes induced by Managed Aquifer Recharge activities: Linking hydrogeological and biological processes"

_Hydrology and Earth System Sciences, 2018_

## Referee Comment (RC1) · 24 May 2018

This study integrated different fields such as hydrogeology, ecology and microbiology applied to a real MAR facility, relating flow (infiltration) conditions, physicochemical water parameters, and microbial changes induced by managed recharge along vertical transect. Except for some marking errors, the scientific results and conclusions presented are clear, concise, and well-structured.

---

## Referee Comment (RC2) · Anonymous Referee #2 · 4 Jun 2018

This paper studies an important problem of microbial community change under conditions of Managed Aquifer Recharges (MAR). Choosing a MAR facility located in NE Spain, the authors took water and soil samples during recharge and no recharge periods to compare the microbial community conditions. The authors reached a conclusion that the MAR is a naturally, passive, and efficient technique for biodegradation of pollutants in groundwater. I believe this work provides useful insights to improve our understanding of MAR in biochemical ways. I still have some comments:

1. Does the types of water change with wet/dry scenarios (eg., Type 1water)? This seems possible.

[Figure]

2. I believe the results of microbial communities in groundwater (Results section 3.1) should also be discussed in the Discussion section. Some of the results are lack of deep interpretation or further discussions.

3. Can results and conclusions of this research be extended to other areas?

---

## Author Comment (AC1) · 12 Jun 2018

We thank Mr. Sang for his comment. We are very pleased our work has been evaluated so positively. We really hope our multidisciplinary paper will be useful for many professionals from different fields of expertise. Any further comment or correction will be welcome.

---

## Author Comment (AC2) · 13 Jun 2018

We want to acknowledge Anonymous Referee 2 for his/her evaluation of the present work. Regarding the comments of the Referee, we would like to clarify some aspects that maybe are not fully-developed in the manuscript.

Referee comment 1: Does the types of water change with wet/dry scenarios (eg., Type 1 water)? This seems possible.

We expect that background conditions (represented by Type I water) may not affect too much the microbial composition in comparison with the conditions induced by recharge

process. One of the evidences that supports this approach is the fact that diversity indices (mainly Shannon and Evenness) remains constant in Type I waters, and decrease accordingly to the influence of the recharge process in wet scenario. Under this assumption, we divide the samples according to the influence that they receive from the recharge process. For doing this, we relied on the information provided by temperature and conductivity profiles (Figures 2 and 3) and the knowledge provided by other studies performed in the same study zone (Valhondo et al., 2014, 2018).

Referee comment 2: I believe the results of microbial communities in groundwater (Results section 3.1) should also be discussed in the Discussion section. Some of the results are lack of deep interpretation or further discussions.

In our opinion, the fitting of our results in the Intermediate Disturbance Hypothesis is wide-argued in the discussion section. Furthermore, we discuss the potential role of some microbial phylotypes related to aerobic oxidation, dehalogenation and denitrification processes. Likewise, we relate such presence with the influence received of recharge process. However, it is true that the lack of information about the contaminants presents in the system do not allow us to conclude the empirical relationship between pollutant concentrations and the presence of these phylotypes. In this way, we propose to include paragraph explaining this lack of evidence in order to propose future research work.

Referee comment 3: Can results and conclusions of this research be extended to other areas?

Yes, some of them can. Of course, we present results from a field experiment, that intrinsically is not performed in controlled conditions. We think that field studies are indeed realistic but their transferability becomes more difficult. However, here we present some interpretations of our study that could be transferred to other sites/experiments: a) The diversity indices can be perfectly compared between sites, whatever it were the approach (laboratory, pilot or field studies). The methodology that we followed for the

calculation of such indexes is well-reported and widely applied. b) The Intermediate Disturbance Hypothesis can be also transferred. In fact, this study evidences that this ecological hypothesis has been accomplished in the Llobregat MAR site. Why could not this approach be applied or accomplished to other impacted environments? c) We present here the results of sequencing tasks that associate some microbial species with certain environmental conditions. We want that our findings would be useful for future research in groundwater microbiology, especially in impacted zones as Llobregat River Basin. Overall, we expect our study it could be transferable to groundwater-surface interaction zones with low nutrient concentration but high amount of micropollutants, emergent organic compounds, etc (as in the case of Llobregat River or other Mediterranean high impacted rivers).

---

## Referee Report (RR1)

The present study adress the impact of Managed Aquifer Recharge (MAR) activities on the features of the in situ microbial community. The latter having been characterized by analysis of both soils and (surface-ground-)water samples, during recharge and no recharge periods. The MAR facilities is located in the NE of Spain and the infiltered water is taken from the Llobregat River, which suffers from contamination by emerging organic contaminants (ECOs) that are not fully removed during wastewater treatments. In this optic the MAR facility induces a perturbation of groundwater characteristic (e.g., organic matter, dissolved oxygen, temperature, pH) leading to the growth and stimulation of the microbial community, favoring the degradation of ECOs. The main goal of the study is that of quantify the MAR related changes on the microbial community. The Authors provide also a link with ecological principles and potential biogeochemical processes.

I think that the paper is interesting, and I applaud the Authors for facing the challenge at a real-world field site! In my opinion the paper is worth for publication after some minor comments are addressed. Note that some of my comments are dictated by my poor background on biological community characterization methods and interpretation.

**Comment 1**

Section 2.3: Are the groundwater samples taken for the microbial analysis taken at the same location of the samples used for the hydrochemistry analysis? I imagine yes, but please make it clear in the text.

**Comment 2**

Section 2.3: "The first one is Richness ($S$), defined as the proportional number of microbial species present in a sample, i.e., equal to the total number of bands;" would it possible to further elaborate on the meaning of these bands? I am not an expert of the subject so if the Authors think that it is not needed, I agree with it, but a more detailed description of the 'bands' meaning would help to attract the interest of a wider audience!

**Comment 3**

Section 2.3: The Shannon ($H$) and Evenness ($S$) are used to characterize the diversity of the microbial community. I suggest to further elaborate on their meaning right after their introduction since it would not be that clear for reader unfamiliar with entropy and measure like that. Note that I am interpreting index $H$ as the entropy of the 'bands spectrum' (perhaps is not the right wording), $H_{max}$ as the entropy of a uniform bands intensity and $E$ accordingly, e.g., $E \to 0$ the microbial community exhibit a poor variability.

**Comment 4**

Section 3.1.2: "The two green triangles in the center of the plot correspond to groundwater samples from P10." and so? The fact that these two triangles of water type IV are not expected to be 'near' the others is not immediate to me, could please the Authors elaborate further this observation.
Furthermore, would be possible to further describe the meaning of the NMDS, its relationship with the bands and so how to read the empty and full circles? Paraphs in Appendix with other notions related with the microbial characterization or adding some reference. Once again if the Authors think that this is standard notions it would not be necessary.

**Comment 5**
Section 3.1.2: Figure 5 in the text is Figure 6 in the figures list. Please make the piezometers identifiers more readable in the dendrogram.

**Comment 6**
Section 3.1.3: I really liked Figure 6 (note that it is called fig. 5 in the list figure) and how it summarizes the lower variability in the bands for water type II an type III w.r.t. the unaffected waters, i.e., type I and IV.

Would it be possible to add the trend for the dry scenario in order to see how the *H* varies as a function of the recharge v.s. no-recharge conditions?

**Comment 7**
Section 3.2: "Figures 8 and 9 show the relative abundance of bacterial phylotypes at the taxonomical level of classes for surface water and soil samples." Should not be "Figures 8 and 9 show the relative abundance of bacterial phylotypes at the taxonomical level of classes for SOIL and WATER samples"?
"For surface water samples (Figure 9), there was a decreasing gradient in community complexity along the ponds" It would beneficial to add a something like " see the *H,* S and E indices in Table 1" or add them to the Fig. 9 (the same for Fig. 8).

**Comment 8**
Section 4.1: I did not know the Intemediate Distrubance Hypotesis (IDH) and I relay liked its application in the current study.

Line 9 pp 9: "in the latter case, values…" upper case is missing at the beginning of the sentence.

**Comment 9**

I really appreciate the fact of deal with the issues dictated by the scale of the investigated dynamics not via widespread (and so thought as unquestionably reliable at all scales) system of partial differential equations (e.g., Darcy flow with mass continuity, advection-dispersion equation), but trough a solid analysis of the available data (essentially following Information Theory metrics). This is a more common practice in the hydrology community, rather than the hydrogeology community. It remains to be seen how to move from laboratory to field scale according with this approach.